# Is PTEN rs397510595 an Unexpected Guardian in Canine Mammary Neoplasia?

**DOI:** 10.3390/ijms262110654

**Published:** 2025-11-01

**Authors:** Ana Canadas-Sousa, Marta Santos, Patrícia Dias-Pereira

**Affiliations:** 1Department of Pathology and Molecular Immunology, ICBAS—School of Medicine and Biomedical Sciences, University of Porto, 4050-313 Porto, Portugal; pdpereira@icbas.up.pt; 2Associated Laboratory for Green Chemistry (LAQV), REQUIMTE, University of Porto, 4051-401 Porto, Portugal; 3Vasco da Gama Research Centre (CIVG), Department of Veterinary Sciences, Vasco da Gama University School (EUVG), 3020-210 Coimbra, Portugal; 4Cytology and Hematology Diagnostic Services, Laboratory of Histology and Embryology, Department of Microscopy, ICBAS, University of Porto, 4050-313 Porto, Portugal; mssantos@icbas.up.pt; 5Oncology Research, UMIB—Unit for Multidisciplinary Research in Biomedicine, ICBAS, University of Porto, 4050-313 Porto, Portugal

**Keywords:** canine mammary neoplasia, PTEN gene, SNP, vascular invasion, survival

## Abstract

Despite steps having been taken to study the influence of genetic polymorphisms in canine mammary neoplasia, the knowledge of their relevance is still incipient compared to the knowledge of human breast cancer. Among tumor suppressor genes, PTEN plays a pivotal role in carcinogenesis; however, the contribution of its constitutional variants to the biology of canine mammary neoplasia remains poorly understood. This observational study assessed the association between PTEN SNPs rs397510595 and rs397513087, genotyped from peripheral blood, and the clinicopathological features, including survival, in a cohort of 206 female dogs with mammary neoplasia. The minor A allele of rs397510595 was present in 17.5% of the population. Carriers of the variant allele were more frequently diagnosed at a late age ≥ 11 years, displayed a complete absence of vascular invasion, and exhibited significantly longer overall survival (mean 22.2 vs. 19.5 months). The SNP rs397513087 did not show a significant association with clinicopathological features or survival. Our data suggests that SNP rs397510595 of the PTEN gene is a putative protective factor for developing canine mammary neoplasia at an early age and might be used as a biomarker for prognostic assessment in dogs with malignant mammary neoplasia.

## 1. Introduction

Cancer is a complex, multifactorial, and multistep disease arising from a wide range of genetic, epigenetic, and phenotypic alterations that drive the transformation of normal cells into neoplastic cells. Advances in molecular and genetic research have greatly expanded our understanding of human breast cancer, allowing the identification of genetic signatures that correlate with neoplasia progression, therapeutic responses, and clinical outcomes [1,2]. Due to their strong epidemiological, histological, and molecular parallels with human breast cancer, CMNs are recognized as valuable spontaneous models in comparative oncology, providing unique opportunities to investigate shared oncogenic pathways and validate biomarkers relevant for both human and veterinary medicine [3,4,5].

Canine mammary neoplasia (CMN) represents some of the most frequent neoplasms in female dogs, representing up to 70% of all diagnosed neoplasias in this population [6,7]. Their incidence increases with age, peaking between 11 and 13 years, and is strongly influenced by reproductive and hormonal factors [8,9]. Beyond these influences, it has become increasingly evident that host genetic variability contributes significantly to the clinical, histological, and molecular heterogeneity of CMN, influencing not only the risk of neoplasia development but also clinicopathological features and disease progression [10,11,12,13]. Single-nucleotide polymorphisms (SNPs) are the most common form of genetic variation in the genome and can significantly impact cancer susceptibility and tumor biology. In fact, SNPs found within oncogenes, tumor suppressor genes, or genes involved in hormonal signaling and DNA repair pathways may modify protein function or gene expression, as previously demonstrated in human oncology studies [14,15] influencing clinicopathological features and clinical behavior of various neoplasia types [14].

Within this framework, special attention has been directed toward tumor suppressor genes, including the Phosphatase and Tensin homolog (PTEN). PTEN is among the most frequently inactivated tumor suppressor genes in human cancers, consistently ranking second in importance after TP53. The prevalence of its inactivation varies among different neoplasia types, but PTEN is nevertheless considered one of the key genes associated with cancer risk, including breast cancer [16,17,18,19,20]. In female dogs, a previous study did not establish a significant association between PTEN genetic variation and the overall risk of developing CMN [21]. However, in other studies, the relationship between PTEN genetic variations and neoplasia subtype, grade, and proliferative activity of CMN has been described, reporting a significant influence of polymorphisms in this gene and the prognosis and the therapeutic response [4,11,22,23,24,25].

PTEN encodes a dual lipid and protein phosphatase that acts as a major negative regulator of the phosphoinositide 3 kinase (PI3K)/AKT/mechanistic target of rapamycin (mTOR) pathway, with a key role in controlling a wide range of essential cellular processes including cell proliferation, growth, survival, and metabolism [26,27]. PTEN blocks PI3K by dephosphorylating phosphatidylinositol (PI)-3,4,5-triphosphate (PIP3) to PI-4,5-bisphosphate (PIP2), thus counteracting PI3K function and subsequent AKT/mTOR activation [28,29].

In human breast cancer, PTEN is frequently inactivated—reported in up to 30–40% of cases when considering loss of expression, deletions, promoter methylation, or post-transcriptional mechanisms—whereas true mutations are less common (~5–10%). Its loss is strongly associated with aggressive clinicopathological features, therapy resistance, and shortened survival [30,31,32]. The canine PTEN gene shares over 95% sequence homology with its human counterpart [33,34]. Immunohistochemical studies in canine species have demonstrated PTEN loss in approximately 30% of CMN, correlating with malignancy and poor prognosis [34,35,36,37]. Beyond this, variations in germline PTEN in CMN remain largely unexplored.

To uncover the role of PTEN genetic variation in the biology and prognosis of canine mammary neoplasia (CMN), this study aimed to investigate the association between two PTEN SNPs and the clinicopathological characteristics and clinical outcome of CMN.

## 2. Results

This cohort included 206 bitches with a mean age of 10.1 years (range, 7–18 years old). Only benign neoplasia was diagnosed in 73 female dogs (mean age 9.7 years), while 133 female dogs presented at least one malignant neoplasia (mean age 10.4 years). Mixed benign neoplasia and complex carcinomas were the most frequently diagnosed subtypes, representing 45.2% and 15.8% of the neoplasias, respectively. Multiple neoplasias were diagnosed in 65.5% of the cases. The mean neoplasia size for benign neoplasias was 2.1 cm (range, 0.3 to 8.5 cm diameter), and for malignant neoplasias, 3.7 cm (range, 0.3 to 16.5 cm diameter). Based on the histological grading method 37/106 (34.9%) carcinomas were grade I, 46/106 (43.4%) were grade II, and 23/106 (21.7%) were grade III. The histological grade was not assigned for in situ carcinoma, carcinosarcomas, and sarcomas. Vascular invasion (assessed in 130 malignant cases) and lymph node metastasis (lymph nodes were collected during the macroscopic examination of the mammary specimens in 116 cases) were observed in 22/130 (16.9%) and in 31/116 (26.7%) of cases, respectively. Two-year follow-up data were available for 116 dogs with at least one malignant neoplasia. Of those, 57/116 (49.1%) were alive at the end of the follow-up period, while 31/116 (26.7%) died due to progression of the disease. Animals lost to follow-up and animals that died from causes not related to the mammary neoplasia were censored (n = 28/116; 24.1%).

### 2.1. SNP rs397510595

Of 206 dogs, for PTEN rs397510595, genotype distribution was GG = 170 (82.5%), GA = 32 (15.5%), and AA = 4 (1.9%), corresponding to a minor allele frequency (MAF) of 0.097 (Table 1). Genotype frequencies followed the Hardy–Weinberg equilibrium (*p* > 0.05). Given the rarity of the AA genotype (n = 4), genotype-specific comparisons were underpowered, and expected cell frequencies were below recommended thresholds for standard tests. To preserve statistical validity, association analyses were performed under a dominant model (GA + AA vs. GG).

The significant associations between the rs397510595 polymorphism and clinicopathological parameters are presented in Table 2.

The analysis of age at neoplasia presentation showed a significant link between the rs397510595 genotype and age distribution. Among female dogs developing neoplasias before 11 years of age, 86% were GG compared to 14% who carried the A allele (OR = 0.42; 95% CI 0.20–0.88; *p* = 0.019).

Vascular invasion was absent in all malignant tumors in female dogs carrying the variant allele, whereas such invasion was observed in neoplasia from 22 of 103 (21.4%) female dogs with the GG (OR = 0.066; 95% CI 0.0039–1.12; *p* = 0.007).

So, the distribution showed a complete segregation of vascular invasion status by genotype. No statistically significant association was observed between the rs397510595 genotype and the neoplasia size, type of growth, histological grade, neoplasia multiplicity (single vs. multiple), benign versus malignant classification, or lymph node metastasis status.

Survival analysis included 116 dogs with malignant neoplasia and complete follow-up data. A-allele carriers demonstrated longer overall survival (mean = 22.2 months; 95% CI 20.3–24.2) compared with dogs with the GG (mean = 19.5 months; 95% CI 17.9–21.1) (log-rank χ^2^ = 3.949; *p* = 0.047) (Figure 1). At the last follow-up, 88.0% of A allele carriers were alive, compared to 69.2% of GG female dogs.

### 2.2. SNP rs397513087

Genotyping of PTEN rs397513087 was successful in 205 out of 206 cases (99.5%). Genotype distribution for PTEN rs397513087 conformed to the Hardy–Weinberg equilibrium. Given the low minor allele frequency and the resulting small number of T-allele carriers (≈10%), the study was underpowered for meaningful association testing with histopathological features or survival outcomes (Table 3). Therefore, we did not pursue subgroup analyses for this SNP.

## 3. Discussion

This study aimed to assess the influence of PTEN genetic variants on clinicopathological features and biological behavior of CMNs. The first step of the study design was to estimate the frequency of the allele variants. While for the SNP rs397510595, the prevalence of the A allele carriers in the cohort was 17%, for the SNP rs397513087, the frequency of variants was below 10%. These findings showed that the wildtype genotype of PTEN was preserved in a large majority of the female dogs included in the cohort.

The low frequency of the minor allele of the SNP rs397513087 jeopardized the assessment of statistically significant associations between the variants and the biology of the CMNs, but this does not completely exclude the possibility of biological effects of the SNP. Even so, the influence of the PTEN SNP rs397513087, if any, in the context of CMNs would be marginal. Considering the future developments and widespread use of genotyping techniques in the management of female dogs with CMNs (personalized medicine) or female dogs at risk of developing CMNs (preventive medicine), it is equally important to report SNPs that had and those that did not have a considerable impact on the context of CMNs. However, given the small number of homozygous (AA) dogs, the estimated effect sizes must be interpreted with caution, as this limitation may have reduced statistical power and widened confidence intervals.

Interestingly, the variant genetic form of SNP rs397510595 (A allele carriers) was more frequently observed in a cohort of female dogs with mammary neoplasias. Female carriers of this genetic variant were diagnosed with mammary neoplasia later in life compared to those with the wild-type GG. The absence of vascular invasion in all A allele carriers is particularly striking, given that this invasion occurs in approximately 25–30% of malignant canine mammary neoplasias and is strongly associated with metastatic spread and poor prognosis [38,39]. It should be emphasized that constitutional SNPs are always part of a broader genetic signature that can influence—mainly indirectly—the onset and course of disease, rather than exerting direct causal effects. Therefore, the association between SNP rs397510595 (A allele carriers) and longer survival should be interpreted within this framework. What is currently known is that PTEN activity interferes with angiogenesis and intravasation, two critical steps in neoplasia progression [40,41]. PTEN exerts its tumor suppressor function largely through antagonism of the PI3K/AKT pathway, thereby downregulating key angiogenic mediators such as VEGF and HIF-1α [42,43]. Experimental models have confirmed that PTEN loss enhances VEGF production and angiogenesis through PI3K/AKT/VEGF/eNOS signaling [44,45], while elevated PTEN levels or optimized splicing constrain this cascade [43,46]. SNP rs397510595 is located in a splice region, suggesting potential functional consequences through altered splicing efficiency or mRNA processing rather than changes in protein activity, given the synonymous nature of the variant. These mechanisms might modulate PTEN expression levels or isoform ratios, strengthening the inhibitory effect on PI3K/AKT-mediated angiogenesis and reducing vascular invasion. Nevertheless, this interpretation remains hypothetical, as no functional validation was performed in this study. Alternatively, this SNP may be in linkage disequilibrium with another functional variant that enhances PTEN activity. Consequently, neoplasia arising in A allele carriers may remain less vascularized, with limited nutrient supply and decreased metastatic potential. Such a cascade would also explain the survival benefit observed in female dogs carrying the variant allele of SNP rs397510595. In human oncology, most PTEN variants are deleterious, and protective effects of specific SNPs have been reported in certain populations. For instance, polymorphisms such as rs701848 and rs2735343 have been associated with variable cancer risk depending on ethnic background [47]. These findings support the notion that germline variants can exert context-dependent modulation of PTEN function. The protective association of rs397510595 in dogs suggests the possibility of species-specific regulatory mechanisms, though cross-species validation should be designed for future studies.

In a comparative oncology context, our results state the path for studying the interplay between PTEN function, angiogenesis, and neoplasia dissemination. Understanding how these variants influence vascular invasion in CMN has a potential translational value for human breast cancer, where suppression of PI3K/AKT-driven angiogenesis remains a major therapeutic objective [41].

Several limitations should be acknowledged. The A allele was rare, and no carriers showed vascular invasion, which required Fisher’s exact test and may have resulted in imprecise estimates. Notably, five A allele carriers developed lymph node metastasis despite no observed vascular invasion, suggesting that lympho-vascular invasion may have occurred in unexamined neoplasia sections. This highlights a key methodological limitation: neoplasia assessment was based on a complete, but still a single representative slab, even for lesions exceeding 5 cm, which may not fully capture intra-neoplasia heterogeneity or detect focal invasion events. Moreover, the study was observational and conducted in a single institution and country, based on a geographically limited canine population, which may not fully represent the genetic diversity of the species, further limiting generalizability. Additionally, pathway markers were not evaluated, preventing validation of the proposed PI3K/AKT/mTOR mechanisms. Given that rs397510595 is located in a splice region with potential direct functional consequences, future studies should prioritize functional validation of splice-regulatory effects alongside molecular pathway analysis. Larger, more diverse canine populations and comparative analyses across breeds are needed to increase statistical power and confirm the impact of this variant on tumor progression and prognosis.

## 4. Materials and Methods

### 4.1. Study Population

The observational study was conducted on 206 female dogs with histologically confirmed mammary neoplasia. Dogs were treated with unilateral mastectomy (removal of an entire unilateral mammary chain) and/or partial mastectomy (removal of one to three mammary gland pairs, including the affected ones). Owners provided consent for surgery with curative intent, as well as for the use of the material for research purposes. This protocol was approved by the Ethics Committee of the Institute of Biomedical Sciences Abel Salazar, University of Porto (ORBEA; P151/2016).

### 4.2. Histopathology

Neoplasia samples were collected and formalin-fixed, paraffin-embedded, and stained with H&E for histological examination. The histological diagnosis was established by a consensus of three pathologists (ACS, MS, and PDP) under a multi-head microscope, according to the official classification criteria for CMNs [48]. Clinicopathological parameters assessed included age, number of neoplasia, biological behavior (neoplasia type), growth pattern, neoplasia size, grade (including parameters such as tubule formation, nuclear pleomorphism, and mitotic count), vascular, and lymph node status. Concerning neoplasia size, the largest diameter measured by the same pathologist (ACS) during trimming was considered and recorded. The neoplasia growth pattern was evaluated and classified as expansive when it was delimited by a capsule, or infiltrative when not delimited by a capsule but without signs of vascular invasion or lymph node metastasis, and invasive if vascular invasion or lymph node metastases were observed. In cases of dogs with multiple malignant neoplasia, a reference lesion was assigned for the statistical analysis of outcomes. The reference lesion was considered as the neoplasia presenting peritumoral vascular invasion (primary criterion), higher nuclear pleomorphism (secondary criterion), or the one with the largest diameter (tertiary criterion) according to a previous study [49]. Histological grading was performed by consensus of three pathologists (ACS, MS, and PDP) based on the assessment of three parameters: tubule formation, nuclear pleomorphism, and mitotic counts [48].

Following Santos et al. (2013), a veterinary adaptation of the human Nottingham Prognostic Index -the Veterinary Canine Prognostic Index (vet-CPI)- was also computed. The vet-CPI was calculated as [neoplasia size (cm) × 0.2] + histological grade (HG; 1, 2, or 3 for grades I, II, and III, respectively) + evidence of vascular invasion/regional lymph node metastases (1 if absent, 2 if present) [49,50,51].

Two-year follow-up data were obtained by consulting the medical records and the referring veterinarian. Disease-specific survival was calculated from the time of diagnosis to the date of the animal’s death/euthanasia due to the progression of the neoplastic disease. Animals that died or were euthanized for unrelated causes and those that were lost to follow-up were censored, respectively, at the time of death and at the time of their last clinical examination. Euthanasia was performed independently by the attending veterinarians, based solely on clinical condition and welfare considerations, and was not related to or influenced by the present study.

### 4.3. Genotyping

Genomic DNA was extracted from peripheral blood. SNP rs397510595 (G > A) was genotyped using validated molecular methods, specifically the High—Purity PCR Template Preparation Kit (Roche, Mannheim, Germany). SNP genotyping was performed using MassARRAY iPLEX Gold Technology at the Unidade de Genómica/Serviço de Genotipagem do Instituto Gulbenkian de Ciência. The protocol for assessing DNA quality and for the recognition of the SNP allele followed previous descriptions [52].

### 4.4. Inclusion and Exclusion Criteria

Only female dogs with histologically confirmed mammary neoplasias, available clinical and follow-up data, and sufficient biological material for genotyping were included. Cases lacking clinicopathological information or suitable DNA samples were excluded from the study.

### 4.5. Statistical Analysis of Data

Two canine PTEN (dog chromosome 26) SNPs were studied: rs397513087, location 26:39440844, a synonymous variant, corresponding to a C/T (Glu-Glu) change—wild-type: CC; variant genotype: T allele carriers; and rs397510595, located at splice region variant 26:39437247, corresponding to a G/A change (Cys-Cys)—wild-type: GG; variant genotype: A allele carriers.

Associations between genotype and categorical variables were analyzed with Pearson’s chi-square or Fisher’s exact test. Odds ratios (OR) with 95% confidence intervals (CI) were calculated. Disease Specific Survival (DSS). The survival analyses (DSS) were performed in cases with at least one malignant neoplasia for which follow-up data were available, using Kaplan–Meier curves and log-rank tests. A 5% level was considered to define statistical significance. Analyses were performed in SPSS v29.9.

## 5. Conclusions

The SNP rs397510595 in the *PTEN* gene appears to have a protective role in female dogs affected by mammary neoplasia. The present data provide the first evidence of a protective PTEN variant in canine oncology, highlighting its potential as a prognostic biomarker and underscoring the importance of investigating both protective and deleterious polymorphisms in veterinary and comparative oncology.

## Figures and Tables

**Figure 1 ijms-26-10654-f001:**
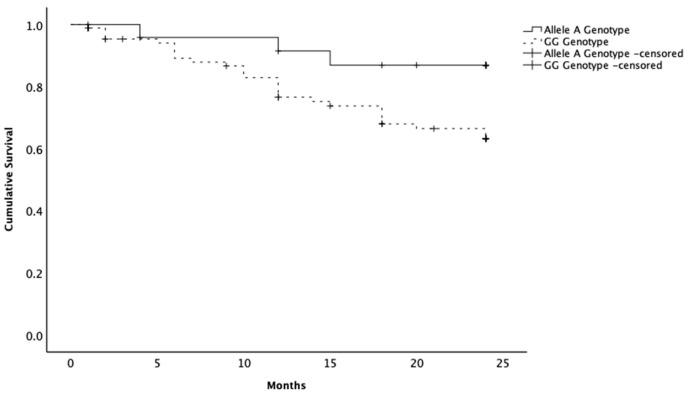
Kaplan–Meier curves of the survival times of female dogs with malignant mammary neoplasia. Animals presenting the allele A variant in the SNP rs397510595 of the PTEN gene showing a longer survival time compared to the female dogs with the wild-type genotype (GG). Censoring is indicated by the + symbol.

**Table 1 ijms-26-10654-t001:** Distribution of genotypes SNP rs397510595 in the PTEN gene in a cohort of 206 female dogs.

Genotype	Frequency	Percentage
AA	4	1.9
GA	32	15.5
GG	170	82.5
Total	206	100.0

**Table 2 ijms-26-10654-t002:** Association between SNP rs397510595 (A/GG allele carriers) PTEN gene with clinicopathological variables in female dogs with mammary neoplasia.

Genotype (n = Number of Cases)			
Independent Variables	A Allele Carriers (n/%)	GG (n/%)	*p*
**Age**			
<11	20 (14.0)	123 (86.0)	
≥11	16 (28.1)	41 (71.9)	***0.019* **
**Number of neoplasia**			
Single	12 (16.9)	59 (83.1)	
Multiple	24 (17.8)	111 (82.2)	NS
**Neoplasia size**			
≤3 cm	20 (17.5)	94 (82.5)	
>3 cm	14 (17.7)	65 (82.3)	NS
**Neoplasia size (only malignant)**			
≤3 cm	13 (20.3)	51 (79.7)	
>3 cm	13 (20.0)	52 (80.0)	NS
**Biological behavior**			
Benign	8 (11.0)	65 (89.0)	
Malignant	28 (21.1)	105 (78.9)	0.068
**Mode of growth pattern**			
Expansive	10 (19.6)	41 (80.4)	
Infiltrative	13 (27.7)	34 (72.3)	
Invasive	5 (14.3)	30 (85.7)	NS
**HG parameters scores**			
**Tubule formation**			
Score 1	4 (16.0)	21 (84.0)	
Score 2	13 (28.3)	33 (71.7)	
Score 3	7 (20.0)	28 (80.0)	NS
**Nuclear pleomorphism**			
Score 1	1 (10.0)	9 (90.0)	
Score 2	18 (26.5)	50 (73.5)	
Score 3	5 (17.9)	23 (82.1)	NS
**Mitotic index**			
Score 1	9 (20.5)	35 (79.5)	
Score 2	5 (19.2)	21 (80.8)	
Score 3	10 (27.8)	26 (72.2)	NS
**HG**			
Grade I	9 (24.3)	28 (75.7)	
Grade II	9 (19.6)	37 (80.4)	
Grade III	6 (26.1)	17 (73.9)	NS
**Vet-CPI**			
≤4.0	15 (24.6)	46 (75.4)	
>4.0	9 (20.0)	36 (80.0)	NS
**Vascular invasion**			
No	27 (25.0)	81 (75.0)	
Yes	0 (0.0)	22 (100.0)	***0.007* **
**Lymph node metastases**			
No	19 (22.4)	66 (77.6)	
Yes	5 (16.1)	26 (83.9)	NS

Legend: SNP: single-nucleotide polymorphism; HG: histological grade; vet-CPI: veterinary canine prognostic index; NS: Not Significant.

**Table 3 ijms-26-10654-t003:** Distribution of genotypes SNP rs397513087 of the PTEN gene in a cohort of 206 female dogs.

Genotype	Frequency	Percentage
CC	186	90.7
CT	18	8.8
TT	1	0.5
Total	205	100.0

## Data Availability

All relevant data are contained within the article, and additional information can be provided by the corresponding author upon reasonable request.

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
