# Peer review of "Is PTEN rs397510595 an Unexpected Guardian in Canine Mammary Neoplasia?"

_ijms, 2025, doi:10.3390/ijms262110654_

Round 1
Reviewer 1 Report
Comments and Suggestions for Authors
Dear authors,
Some consideration is presented above regarding the article, a quite interesting matter considering the frequency of the alteration in bitches over the world.
- Many technical terms are used, nowadays, to describe the alteration which is matter of this study; although there are many discussion related to the topic, it is important to consider the basic concepts of pathology to name the alteration; the term “tumor” is a general one, which describes an increasing volume in a patient; differently, neoplasm is described as an “increasing volume” provoked by an alteration identified at cytology, named neoplasia; some literature bring this distinction (Withrow & MacEwen’s Small Animal Clinical Oncology, 2020). In this sense, consider adjust the general term “tumor” in the text and title by neoplasia or neoplasm, considering is the alteration is micro or macroscopic; the suggestion is contrary to a large part of the available studies, but considering the technical presentation, the change is important;
- Lines 57-58: is presented a list of references, but it is not mentioned the target specie of the information. Consider include the specie which the studies were performed, considering the paragraph starts with the description in dogs;
- Line 59: remove the “.” After types;
- Lines 89-91: Is the information related to canine? Consider clarify the information in the text;
- The section material and methods are presented after the discussion; please, make the adjustment in this order;
- The use of table can be interesting to show some data presented at “2. Results”, which can improve understanding;
- Lines 176-204: consider reduce the paragraph size and long phrases;
- Along the text, words as tumors, cancer and neoplasia were used, apparently, as synonymous; consider verify and standardize the terms, if possible;
- Lines 205-206: “In human oncology, most PTEN variants are deleterious, and protective effects of specific SNPs have been reported in certain populations.”; in this sense, consider include the study limitations, considering the dog population included;
- Line 235: consider use the surgical term “unilateral mastectomy”; surgically, radical include removing of “both mammary chains”;
- Lines 232: inclusion/exclusion criteria were not presented; is the absence of some information considered to determine exclusion?
- Line 272: consider include that euthanasia was performed with no relation with this study; the present information is vague, appears not correlated with the study.
My best regards,
Author Response
IJMS (ISSN 1422-0067)
Manuscript ID ijms-3943003
Title: Is PTEN rs397510595 an Unexpected Guardian in Canine Mammary Tumors?
Authors Ana Canadas-Sousa * , Marta Santos , Patrícia Dias-Pereira
Section Molecular Genetics and Genomics
Special Issue Role of Mutations and Polymorphisms in Various Diseases: 2nd Edition
The authors would like to express their gratitude to the reviewers for their careful reading of the manuscript. We greatly appreciate the time and effort dedicated to the review process, as well as the constructive feedback provided, which undoubtedly contributed to improving the overall quality and presentation of our manuscript.
We have carefully revised our manuscript in light of the suggestions and comments receives, and have addressed each point in detail. Our responses to each individual comment are provided in the following section.
Reviewer 1
Some consideration is presented above regarding the article, a quite interesting matter considering the frequency of the alteration in bitches over the world.
Many technical terms are used, nowadays, to describe the alteration which is matter of this study; although there are many discussion related to the topic, it is important to consider the basic concepts of pathology to name the alteration; the term “tumor” is a general one, which describes an increasing volume in a patient; differently, neoplasm is described as an “increasing volume” provoked by an alteration identified at cytology, named neoplasia; some literature bring this distinction (Withrow & MacEwen’s Small Animal Clinical Oncology, 2020). In this sense, consider adjust the general term “tumor” in the text and title by neoplasia or neoplasm, considering is the alteration is micro or macroscopic; the suggestion is contrary to a large part of the available studies, but considering the technical presentation, the change is important;
A: Thank you very much for this thoughtful and constructive comment. The authors truly appreciate the reviewer’s clarification regarding the technical distinction between the terms tumor and neoplasia/neoplasm. We fully agree that adopting the more precise terminology improves the scientific rigor and accuracy of the manuscript. Accordingly, we will revise the text and title to replace “tumor” with “neoplasia” where appropriate, in alignment with the reviewer’s valuable suggestion.
Lines 57-58: is presented a list of references, but it is not mentioned the target specie of the information. Consider include the specie which the studies were performed, considering the paragraph starts with the description in dogs;
Thank you for this helpful comment. We agree with the reviewer’s observation and have revised the paragraph accordingly. We added the specification that the cited studies refer to human oncology studies, as follows:
“In fact, SNPs found within oncogenes, tumor suppressor genes, or genes involved in hormonal signaling and DNA repair pathways may modify protein function or gene expression, as previously demonstrated in human oncology studies (Li et al., 2002; Liu et al., 2014; Yang et al., 2010; Deng et al., 2017; Erichsen & Chanock, 2004), influencing the clinicopathological features and biological behavior of various neoplasia types.”
This modification clarifies the target species and aligns with the reviewer’s suggestion.
Line 59: remove the “.” After types;
Thank you for this careful observation. The extra period after “types” has been removed, as suggested.
Lines 89-91: Is the information related to canine? Consider clarify the information in the text;
Thank you for this helpful comment. We agree with the reviewer and have clarified in the text that the reported data refer to canine mammary neoplasia, as follows:
“Immunohistochemical studies in canine species have demonstrated PTEN loss in approximately 30% of cases, correlating with malignancy and poor prognosis (Asproni et al., 2015; Molín et al., 2024; Qiu et al., 2008; Ressel et al., 2009).”
This adjustment ensures clarity regarding the species and aligns with the reviewer’s suggestion.
The section material and methods are presented after the discussion; please, make the adjustment in this order;
Thank you for this comment. We understand the reviewer’s observation; however, the current structure — with the “Material and Methods” section placed after the “Discussion” — follows the official formatting guidelines of the journal, which specify this order. Therefore, the section has been kept in accordance with the journal’s submission requirements.
The use of table can be interesting to show some data presented at “2. Results”, which can improve understanding;
Thank you for this helpful suggestion. We appreciate the reviewer’s consideration regarding the presentation of these data. However, since this section provides a concise descriptive summary of the study population and the numerical information is already clearly presented in the text, the authors opted to maintain it in paragraph form. This approach ensures a smoother narrative flow and avoids unnecessary redundancy with the analytical tables presented later in the manuscript.
Lines 176-204: consider reduce the paragraph size and long phrases;
Thank you for this valuable suggestion. The paragraph has been revised to improve clarity, fluency, and scientific accuracy, following the reviewer’s recommendation. We believe that the updated version provides a more concise and coherent discussion of the findings while maintaining all relevant details and references.
“Interestingly, the variant genetic form of SNP rs397510595 (A allele carriers) was more frequently observed in a cohort of female dogs with mammary neoplasias. Female carriers of this genetic variant were diagnosed with mammary neoplasia later in life compared to those with the wild-type GG genotype. The absence of vascular invasion in all A allele carriers is particularly striking, given that this invasion occurs in approximately 25-30% of malignant canine mammary neoplasias and is strongly associated with metastatic spread and poor prognosis (Diessler et al., 2017; Pastor et al., 2020). It should be emphasized that constitutional SNPs are always part of a broader genetic signature that can influence—mainly indirectly—the onset and course of disease, rather than exerting direct causal effects. Therefore, the association between SNP rs397510595 (A allele carriers) and longer survival should be interpreted within this framework. What is currently known is that PTEN activity interferes with angiogenesis and intravasation, two critical steps in neoplasia progression (Feng et al., 2020; Wen et al., 2001). PTEN exerts its tumor suppressor function largely through antagonism of the PI3K/AKT pathway, thereby downregulating key angiogenic mediators such as VEGF and HIF-1α (Jiang & Liu, 2009; Tamayo et al., 2023). Experimental models have confirmed that PTEN loss enhances VEGF production and angiogenesis through PI3K/AKT/VEGF/eNOS signaling (Huang & Kontos, 2002; Ma et al., 2009), while elevated PTEN levels or optimized splicing constrain this cascade (Karar & Maity, 2011; Tamayo et al., 2023). SNP rs397510595 is located in a splice region, suggesting potential functional consequences through altered splicing efficiency or mRNA processing rather than changes in protein activity, given the synonymous nature of the variant. These mechanisms might modulate PTEN expression levels or isoform ratios, strengthening the inhibitory effect on PI3K/AKT-mediated angiogenesis and reducing vascular invasion. Alternatively, this SNP may be in linkage disequilibrium with another functional variant that enhances PTEN activity. Consequently, neoplasias arising in A allele carriers may remain less vascularized, with limited nutrient supply and decreased metastatic potential. Such a cascade would also explain the survival benefit observed in female dogs carrying the variant allele of SNP rs397510595.”
Along the text, words as tumors, cancer and neoplasia were used, apparently, as synonymous; consider verify and standardize the terms, if possible;
Thank you for this helpful comment. We agree with the reviewer’s observation and have carefully reviewed the manuscript to ensure consistent terminology. The terms have been standardized throughout the text, using “neoplasia” as the preferred term, in accordance with pathological nomenclature.
Lines 205-206: “In human oncology, most PTEN variants are deleterious, and protective effects of specific SNPs have been reported in certain populations.”; in this sense, consider include the study limitations, considering the dog population included;
Thank you for this valuable comment. We agree with the reviewer’s suggestion and have incorporated an additional statement in the limitations paragraph, clarifying that the study population was restricted to a specific geographical region and may not represent the full genetic diversity of the canine species. This adjustment addresses the limitation related to the dog population included.
“Moreover, the study was observational and conducted in a single institution and country, based on a geographically limited canine population, which may not fully represent the genetic diversity of the species, further limiting generalizability.”
Line 235: consider use the surgical term “unilateral mastectomy”; surgically, radical include removing of “both mammary chains”;
Thank you for this helpful observation. We agree with the reviewer and have replaced the term “radical mastectomy” with “unilateral mastectomy” in the text, as it accurately describes the surgical procedure performed in the dogs included in this study.
Lines 232: inclusion/exclusion criteria were not presented; is the absence of some information considered to determine exclusion?
Thank you for this valuable comment. We agree with the reviewer and have added a brief description of the inclusion and exclusion criteria in the Study Population section, specifying that only cases with complete clinicopathological, molecular, and follow-up data were included, and those with missing information were excluded from the study.
“Inclusion and exclusion criteria
Only female dogs with histologically confirmed mammary neoplasias, available clinical and follow-up data, and sufficient biological material for genotyping were included. Cases lacking clinicopathological information or suitable DNA samples were excluded from the study.”
Line 272: consider include that euthanasia was performed with no relation with this study; the present information is vague, appears not correlated with the study.
Thank you for this helpful comment. We agree with the reviewer’s observation and have included an additional statement clarifying that euthanasia was performed independently of this study, based solely on clinical judgment and animal welfare considerations. This addition ensures that the information is unambiguous and ethically clear.
“Euthanasia was performed independently by the attending veterinarians, based solely on clinical condition and welfare considerations, and was not related to or influenced by the present study.”
Reviewer 2 Report
Comments and Suggestions for Authors
1- the small number of homozygous (AA) genotype is a significant limitation although the dominant model (GA+AA vs. GG) is a reasonable approach, it would be valuable to include more detailed discussion of the effect sizes.
2- The discussion presents a compelling hypothesis that the rs397510595 SNP may affect PTEN function through altered mRNA processing. However, this is speculative as no functional data are provided.
some minors
Please ensure consistency in statistical notation throughout
the abstract should clearly specify that this was an observational, retrospective study.
The manuscript is well-written. However, some grammatical polishing is suggested in a few places
for example, "female dogs’ carriers" could be rephrased to "female dogs carrying the variant allele" or "A-allele carriers."
Author Response
IJMS (ISSN 1422-0067)
Manuscript ID ijms-3943003
Title: Is PTEN rs397510595 an Unexpected Guardian in Canine Mammary Tumors?
Authors Ana Canadas-Sousa * , Marta Santos , Patrícia Dias-Pereira
Section Molecular Genetics and Genomics
Special Issue Role of Mutations and Polymorphisms in Various Diseases: 2nd Edition
The authors would like to express their gratitude to the reviewers for their careful reading of the manuscript. We greatly appreciate the time and effort dedicated to the review process, as well as the constructive feedback provided, which undoubtedly contributed to improving the overall quality and presentation of our manuscript.
We have carefully revised our manuscript in light of the suggestions and comments receives, and have addressed each point in detail. Our responses to each individual comment are provided in the following section.
Reviewer 2
1- the small number of homozygous (AA) genotype is a significant limitation although the dominant model (GA+AA vs. GG) is a reasonable approach, it would be valuable to include more detailed discussion of the effect sizes.
Thank you for this valuable comment. We agree with the reviewer’s observation and have added a clarifying statement in the discussion acknowledging that, due to the small number of homozygous (AA) dogs, the estimated effect sizes should be interpreted with caution, as this limitation may have reduced the statistical power and precision of the results.
“This study aimed to assess the influence of PTEN genetic variants on clinicopathological features and biological behavior of CMNs. The first step of the study design was to estimate the frequency of the allele variants. While for the SNP rs397510595, the prevalence of the A allele carriers in the cohort was 17%, for the SNP rs397513087, the frequency of variants was below 10%. These findings showed that the wildtype genotype of PTEN was preserved in a large majority of the female dogs included in the cohort.
The low frequency of the minor allele of the SNP rs397513087 jeopardized the assessment of statistically significant associations between the variants and the biology of the CMNs, but this does not completely exclude the possibility of biological effects of the SNP. Even so, the influence of the PTEN SNP rs397513087, if any, in the context of CMNs would be marginal. Considering the future developments and widespread use of genotyping techniques in the management of female dogs with CMNs (personalized medicine) or female dogs at risk of developing CMNs (preventive medicine), it is equally important to report SNPs that had and those that had not a considerable impact in the context of CMNs. However, given the small number of homozygous (AA) dogs, the estimated effect sizes must be interpreted with caution, as this limitation may have reduced statistical power and widened confidence intervals.”
2- The discussion presents a compelling hypothesis that the rs397510595 SNP may affect PTEN function through altered mRNA processing. However, this is speculative as no functional data are provided.
Thank you for this insightful comment. We agree with the reviewer and have added a clarifying statement after the description of the proposed mechanism to specify that this interpretation remains hypothetical, as no functional validation was performed in this study.
“...reducing vascular invasion. Nevertheless, this interpretation remains hypothetical, as no functional validation was performed in this study. Alternatively,...”
some minors
Please ensure consistency in statistical notation throughout
the abstract should clearly specify that this was an observational, retrospective study.
Thank you for this helpful observation. The presentation of the statistical results has been carefully reviewed and standardized throughout the manuscript to ensure consistency in format and notation. The relevant section was revised to maintain uniform reporting of statistical parameters (OR, 95% CI, and p-values) and to improve overall clarity and readability, without altering the results or their interpretation.
Comments on the Quality of English Language
The manuscript is well-written. However, some grammatical polishing is suggested in a few places
for example, "female dogs’ carriers" could be rephrased to "female dogs carrying the variant allele" or "A-allele carriers."
Thank you for this kind comment and for the helpful language suggestion. The manuscript has been carefully reviewed, and the phrasing has been revised accordingly (e.g., “female dogs’ carriers” was corrected to “female dogs carrying the variant allele”). Minor grammatical adjustments were also made throughout the text to further improve clarity and fluency.